# Bacterially synthesized tellurium nanostructures for broadband ultrafast nonlinear optical applications

Kangpeng Wang[1,2], Xiaoyan Zhang[1], Ivan M. Kislyakov [1], Ningning Dong[1], Saifeng Zhang[1], Gaozhong Wang[1,3], Jintai Fan[1], Xiao Zou[4], Juan Du[4], Yuxin Leng[4], Quanzhong Zhao[4], Kan Wu[5], Jianping Chen[5], Shaun M. Baesman[6], Kang-Shyang Liao[7], Surendra Maharjan[7], Hongzhou Zhang[3], Long Zhang[1,4,8], Seamus A. Curran[7], Ronald S. Oremland[6], Werner J. Blau [3] & Jun Wang[1,4,8]

Elementary tellurium is currently of great interest as an element with potential promise in nano-technology applications because of the recent discovery regarding its three two-dimensional phases and the existence of Weyl nodes around its Femi level. Here, we report on the unique nano-photonic properties of elemental tellurium particles [Te(0)], as harvest from a culture of a tellurium-oxyanion respiring bacteria. The bacterially-formed nano-crystals prove effective in the photonic applications tested compared to the chemically-formed nano-materials, suggesting a unique and environmentally friendly route of synthesis. Nonlinear optical measurements of this material reveal the strong saturable absorption and nonlinear optical extinctions induced by Mie scattering over broad temporal and wavelength ranges. In both cases, Te-nanoparticles exhibit superior optical nonlinearity compared to graphene. We demonstrate that biological tellurium can be used for a variety of photonic applications which include their proof-of-concept for employment as ultrafast mode-lockers and all-optical switches.

[1] Laboratory of Micro-Nano Optoelectronic Materials and Devices, Laboratory of Laser and Infrared Materials, Key Laboratory of Materials for High-Power Laser, Shanghai Institute of Optics and Fine Mechanics, Chinese Academy of Sciences, Shanghai 201800, China. [2] Department of Electrical Engineering, Technion-Israel Institute of Technology, Haifa 3200003, Israel. [3] School of Physics, CRANN and AMBER Research Centres, Trinity College Dublin, Dublin 2, Ireland. [4] State Key Laboratory of High Field Laser Physics, CAS Center for Excellence in Ultra-intense Laser Science, Shanghai Institute of Optics and Fine Mechanics, Chinese Academy of Sciences, Shanghai 201800, China. [5] State Key Laboratory of Advanced Optical Communication Systems and Networks, Department of Electronic Engineering, Shanghai Jiao Tong University, Shanghai 200240, China. [6] US Geological Survey, Menlo Park, CA 94025, USA. [7] Institute for NanoEnergy, Department of Physics, University of Houston, Houston, TX 77204, USA. [8] Center of Materials Science and Optoelectronics Engineering, University of Chinese Academy of Sciences, Beijing 100049, China. Correspondence and requests for materials should be addressed to K.W. (email: kanwu@sjtu.edu.cn) or to L.Z. (email: lzhang@siom.ac.cn) or to J.W. (email: jwang@siom.ac.cn)

Low dimensional materials play increasingly powerful roles in the electronic and photonic industries[1]. Elemental tellurium [Te(0)] has been recently highlighted because of its unconventional properties[2–5]. Bulk trigonal tellurium has been predicted to have multiple Weyl nodes near its Fermi level[5], opening the possibilities for high carrier-mobility and topological devices. Recently, three two-dimensional tellurene phases (α-, β-, and γ-Te) have been reported[2,3,6,7]. They exhibited excellent carrier-mobility which are 2–3 orders of magnitude higher than that of MoS₂, the most intensively studied 2D analog[3]. In particular, a few layers of α-tellurium can be formed from trigonal Te via a spontaneous phase transition[3,8]. These properties make tellurium a promising candidate as the material of choice for next-generation optoelectronic and photonic devices[2].

The conventional synthesis of tellurium nanomaterials relies heavily on chemical approaches that employ harsh reagents, high temperatures, and high costs associated with considerable hazardous waste disposal. The use of specific anaerobic bacteria that can process the oxyanions of Group 6 metalloids (for example, Se and Te) as respiratory electron acceptors, results in their reduction to the elemental state, allowing their formation at room temperature. This approach has significant applicability in nanophotonics, as it opens the possibility of a non-polluting alternative to more commonly used chemical techniques[9–13]. Furthermore, the biological synthesis route was recently demonstrated for self-assembling electronic devices[14]. However, the physical properties of biologically formed Te(0)-nanoparticles are largely unexplored.

In this work, we harvest elemental tellurium nanostructures formed by cultivating the anaerobic bacterium *Bacillus selenitireducens* with Te(IV) as its electron acceptor[9,10,15]. Nonlinear optical properties of these microbiologically synthesized tellurium (Bio-Te) nanostructures are investigated by open-aperture z-scan over broad temporal and wavelength ranges. We highlight the potential of Bio-Te nanostructures in photonic applications by building ultrafast infrared 1.5 μm fiber and 2 μm solid-state lasers using Bio-Te as the saturable absorber. With mode-locking and Q-switching achieved by Bio-Te, ultrafast pulse generation is observed in these lasers. In addition, an all-optical switch based on Bio-Te is demonstrated for optical fiber systems. Our results suggest that biological Te nanocrystals have the potential for a broad range of photonic applications, such as in ultrafast mid-infrared lasers, and optical routing.

## Results

### Fabrications and characterizations of biological tellurium.

The elemental tellurium nanocrystals were produced by growing Te-oxyanion respiring bacteria and by harvesting the crystals after cultivation as shown in Fig. 1a (see Methods)[10–12,16]. The harvested Bio-Te nanostructures were aggregated into micro-pellets. However, the dense aggregations were unfavorable for the linear and nonlinear optical studies. We, therefore, employed poly(*m*-phenylenevinylene)-*co*-2,5-dioctoxy-phenylenevinylene (PmPV) to disperse the Te(0), thereby forming a nanocomposite[15] (see Supplementary Note 1). PmPV has been proven to be an ideal π-electron-rich host for many types of filler including single-walled carbon nanotubes[17]. The coiled conformation of PmPV enables it to wrap around the Bio-Te nanocrystals (Fig. 1b) which allows sufficiently close intermolecular proximity for π–π interaction to occur. Figure 1c shows the effective dispersion of Bio-Te aggregates of Bio-Te in toluene with the addition of PmPV.

Figure 1d shows sample images obtained with a high-resolution transmission electron microscope (HRTEM) and fast Fourier transformation of Bio-Te dispersed in PmPV. Lattice spacings of 0.32 nm and 0.59 nm were observed corresponding to

the (101) and (001) crystal planes, respectively. This implies that Bio-Te has a triagonal crystal geometry comprised of three Te-atoms per unit cell. The Te-atoms are covalently bonded, forming a series of helical chains[18] that interact with each other by van der Waals forces. We also found that the applied PmPV layer surrounds the crystalline Te, as shown in Fig. 1d. Raman spectroscopy was employed to confirm the atomic structural arrangement of Bio-Te. In Fig. 1e, the Raman spectra of both the pristine Bio-Te and the Bio-Te-PmPV composite exhibited distinct fingerprints in the crystalline trigonal Te[19]. The characteristic bands located at 88.6, 113.6, and 134.9 cm⁻¹ are the first order vibration modes of bending and stretching in the Te crystal. The band observed at 261.9 cm⁻¹ is the second-order Raman vibration. Other peaks at 219.8, 438.4, 574, 640.9, and 657.6 cm⁻¹ belong to TeO₂, which may have originated from a small amount of oxidation of Te(0) through exposure to ambient air over time. We also detected characteristic bands associated with PmPV in the Raman spectrum of the Bio-Te-PmPV, as indicated in Fig. 1e. Linear optical absorption of Bio-Te-PmPV composite is shown in Fig. 1f. This spectrum was achieved by subtracting the PmPV background from the spectrum of the Bio-Te-PmPV composite. A typical broad absorption band centered at 1.9 eV (~650 nm) is observed, which is caused by the transitions from the *p*-nonbonding valence band triplet to the *p*-antibonding conduction band triplet[20,21].

The steady-state photoluminescence (PL) spectra of the Bio-Te-PmPV and PmPV are shown on the right-hand side of Fig. 1f. The PL envelopes of both samples were quite similar, showing intrinsic PLs with a broad peak between 450 nm and 650 nm. This similarity indicates that PmPV molecules retained their PL, whereas the Te crystal did not exhibit PL. However, the PL intensity of the Bio-Te-PmPV was approximately one order of magnitude lower than that of the pure PmPV under the same excitation, suggesting that the quench of the luminescence of PmPV was caused by the Bio-Te nanocrystal. To gain a comprehensive understanding of this effect, we carried out PL decay kinetics measurements of both samples. As shown in Fig. 1g, the PL decay of Bio-Te-PmPV was faster than that of pure PmPV for the first few ns, as marked by the blue shaded region (see inset). We employed a tri-exponential model to fit the PL decay curves, and the results are shown in Supplementary Table 1. The two dominant PL decay lifetimes were fitted to be $\tau_1$ ~338 ps and $\tau_2$ ~1.09 ns for Bio-Te-PmPV, while $\tau_1$ ~ 460 ps and $\tau_2$ ~1.11 ns were observed for PmPV, respectively. The faster PL decay in Bio-Te-PmPV implies a quenching effect caused by by the Bio-Te nanocrystals. A possible explanation for this quenching is the existence of energy transfer from the excited donor PmPV molecules to the acceptor Bio-Te nanocrystals which is evidenced by the overlap of the PmPV PL peak and the Bio-Te absorption band (red shaded area in Fig. 1f). That is, the Te-acceptor nanocrystals consumed the energy of excited carriers and relaxed them to the ground state non-radiatively, leading to the drop of the PL intensity in the first few ns (1–10 ns).

### Nonlinear optical properties of biological tellurium.

To reveal the nonlinear absorptive responses of Bio-Te-PmPV, we carried out a series of open-aperture z-scans with different wavelength excitations using fs pulses (Fig. 2). With the excitation set at 800 nm and applying 200 nJ pulses (maximum of 175 GW·cm⁻²), the Bio-Te-PmPV exhibited saturable absorption (SA), that is, the optical transmission increases with the incident beam intensity when the sample position $z$ approaches the focal point of the focusing lens ($z = 0$ mm) (Fig. 2a). However, the SA response was not observed in PmPV solutions (the squares in Fig. 2a), indicating that the SA response was caused solely by the Te

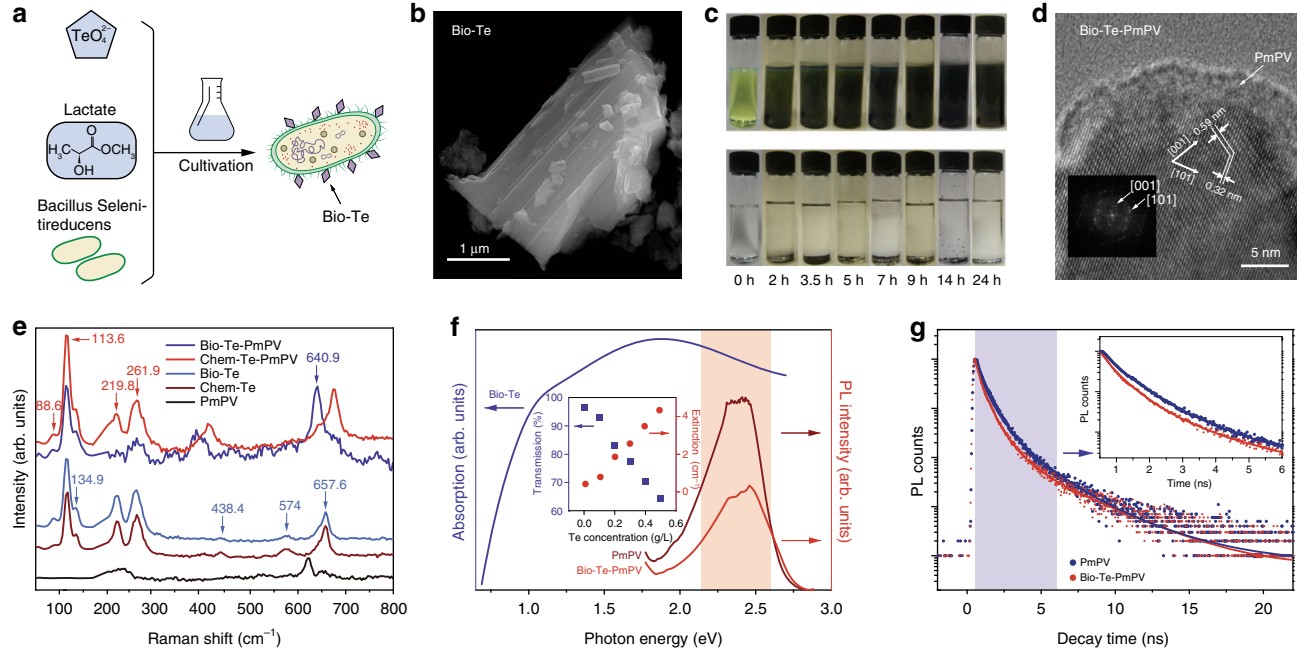

**Fig. 1** Synthesis and characterization of biological synthesized tellurium (Bio-Te). **a** Synthesis scheme of tellurium nanocrystals by anaerobic bacteria, *Bacillus selenitireducens*. **b** Image of Bio-Te crystalline nano-flake taken by scanning transmission electron microscope. **c** Dispersion of Bio-Te in PmPV/ toluene (upper) and in toluene only (bottom) with increasing stirring times on the X-axis, showing the preparation of Bio-Te-PmPV composites. PmPV is abbreviation of poly(*m*-phenylenevinylene)-*co*-2,5-dioctoxy-phenylenevinylene. **d** Transmission electron microscopy image of a Bio-Te crystalline nano-flake wrapped by PmPV layers. Inset is the image after fast Fourier transformation (FFT). **e** Raman spectra of Bio-Te, Bio-Te-PmPV, chemically synthesized tellurium nanocrystals (Chem-Te), Chem-Te-PmPV, and PmPV. **f** Left: the absorption spectrum of Bio-Te. This curve was obtained by subtracting the absorption of PmPV (0.5 g/L in toluene) from that of Bio-Te-PmPV. Right: photoluminescence (PL) spectra of Bio-Te-PmPV and pure PmPV. Inset: optical linear transmission and extinction coefficient as functions of Te concentrations at 532 nm. **g** The PL decay kinetics of PmPV and Bio-Te-PmPV at 528 nm excitation. The inset shows a blow up of the blue shaded region, showing the quenching effect caused by tellurium

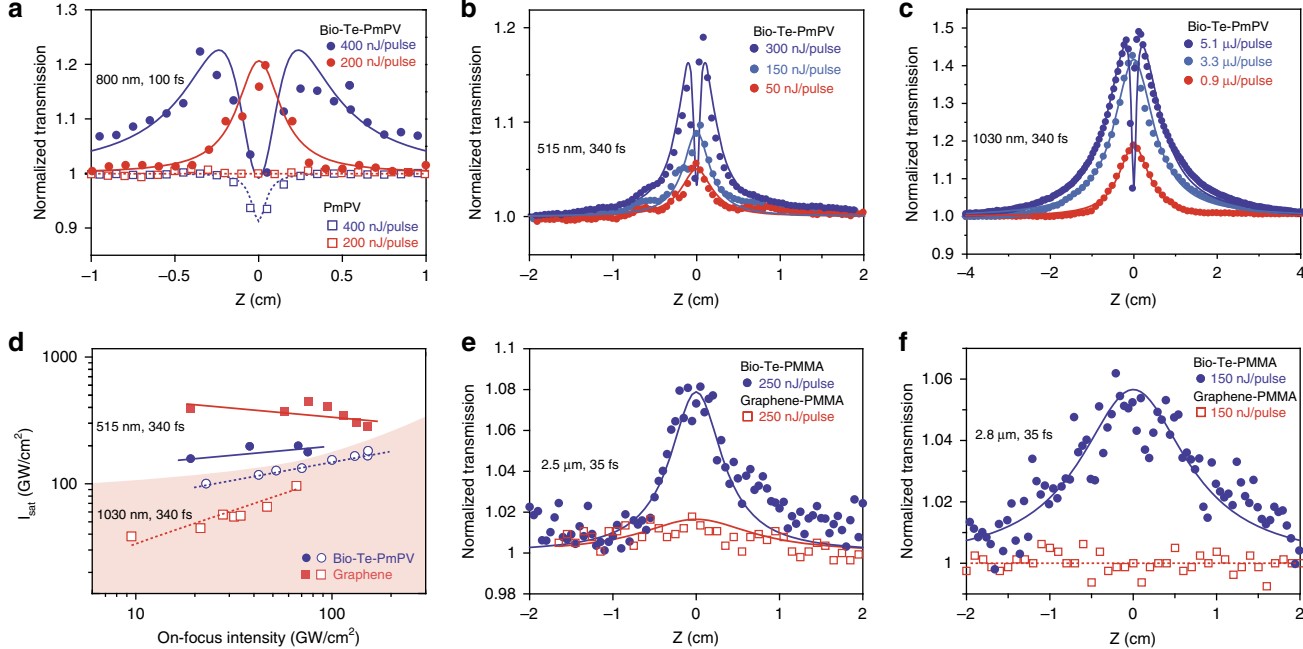

**Fig. 2** Open-aperture z-scan results of the biological synthesized tellurium samples. **a** Experimental results with fs pulses, 800 nm. Bio-Te-PmPV (solid circles); PmPV (hollow squares). Solid lines: fitted z-scan curves using equation (1); Dashed lines are for visual guide. **b**, **c** Experimental results with 340 fs laser at 515 nm (**b**) and 1030 nm (**c**). **d** Saturated intensity $I_{sat}$ of Bio-Te-PmPV and graphene dispersion as functions of the laser intensity at 1030 nm and 515 nm. **e**, **f** Mid-infrared open z-scans of Bio-Te and graphene polymethyl methacrylate (PMMA) films at 2.5 μm and 2.8 μm wavelengths, showing better saturable absorptive responses of Bio-Te than those of graphene

nanocrystals. Because the excitation photon energy (1.55 eV), is much larger than the Te bandgap[5] (0.323 eV), the conduction band of Te can be filled at high photon intensities. This bandfilling leads to Pauli blocking and thereafter saturable absorption. As the excitation pulse energy increased to 400 nJ (maximum of 350 GW·cm$^{-2}$), a dip appeared around the center of the z-scan traces, implying that a combined effect of SA and nonlinear extinction (NLE) was taking place. This NLE effect may be attributed to the multiphoton absorption brought by PmPV[15]. The nonlinear optical (NLO) coefficients of PmPV and Bio-Te can be extracted from z-scan traces using the following photon transport equation:

$$\frac{dI}{dz} = -\frac{\alpha_0 I}{1 + \frac{I}{I_{sat}}} - \beta I^2 \qquad (1)$$

where $\alpha_0$ is the linear absorption coefficient, $I$ is the photon intensity, $z$ is the distance traveled by light in the NLO medium, $I_{sat}$ is the saturated intensity contributed by Bio-Te, and $\beta$ is the NLE coefficient contributed by PmPV. For laser excitations at 515 nm and 800 nm wavelengths, PmPV exhibited a two-photon absorption, therefore $\beta$ can be regarded as the two-photon absorption coefficient ($\beta_{2PA}$) of PmPV. For laser excitation at 1030 nm, PmPV exhibited a three-photon absorption profile, therefore $\beta = \gamma_{3PA} I$, where $\gamma_{3PA}$ is the three-photon absorption coefficient of PmPV. The imaginary part of the third-order NLO susceptibility, $Im\chi^{(3)}$, which is directly related to NLO coefficient $\alpha_{NL}$, is calculated from $Im\chi^{(3)} = [10^{-7} c\lambda n^2/(96\pi^2)]\alpha_{NL}$, where $c$ is the speed of light, $\lambda$ is the excitation wavelength, and $n$ is the refractive index. In the case of Bio-Te, $\alpha_{NL}$ was fully attributed to saturable absorption, that is, $\alpha_{NL} \cong -\alpha_0/I_{sat}$ (Fig. 2). The figure of merit (FOM) of Bio-Te for the third-order optical nonlinearity is defined as FOM $= |Im\chi^{(3)}/\alpha_0|$. The fitted z-scan curves using Eq. (1) are presented in Fig. 2a, and correspond well correspondence with the experimental data obtained for the 200 nJ/pulse to 600 nJ/pulse excitations. The average saturable intensity $I_{sat}$ of Bio-Te at 800 nm was determined to be $261 \pm 176$ GW·cm$^{-2}$. This implies that the SA of Bio-Te was at least comparable to that of black phosphorus (BP) at 800 nm, which was reported as $335 \pm 43$ GW·cm$^{-2}$ [22].

Figure 2b, c show the z-scan traces of Bio-Te-PmPV for 340 fs at 515 nm and 1030 nm laser pulses, which indicate the broadband saturable absorption by Te-nanoparticles over the range of visible to near-infrared (NIR) wavelengths. The linear absorption coefficients ($\alpha_0$) of Bio-Te-PmPV at 1030 and 515 nm were 6.47 cm$^{-1}$ and 6.17 cm$^{-1}$, respectively. For higher intensity laser excitations at 800, 1030, and 515 nm, the multi-photon absorption in PmPV resulted in an obvious turning point in optical transmission around $z = 0$ cm for Bio-Te-PmPV. These

experimental data were fitted using Eq. (1) (Fig. 2b, c). As noted previously at 800 nm, the PmPV only contributed to multiphoton absorption while SA was caused by the Bio-Te. Figure 2d summarizes the saturation intensity ($I_{sat}$) as a function of the on-focus beam intensity for a series of z-scans. In Bio-Te, $I_{sat}$ increased slightly with laser intensity with a median value of $145 \pm 23$ GW·cm$^{-2}$ for 1030 nm, and $201 \pm 35$ GW·cm$^{-2}$ for 515 nm. The median $Im\chi^{(3)}$ and FOM of Bio-Te were respectively determined to be $-(2.76 \pm 0.58) \times 10^{-14}$ esu and $(4.27 \pm 0.91) \times 10^{-15}$ esu·cm for 1030 nm, and $-(1.07 \pm 0.11) \times 10^{-14}$ esu and $(1.74 \pm 0.18) \times 10^{-15}$ esu·cm for 515 nm. The $I_{sat}$ for Bio-Te at 1030 nm was approximately one-third that of BP, implying a much stronger SA response to infrared excitation[23].

We further demonstrated the saturable absorptive response of Bio-Te at mid-infrared wavelengths with ~35 fs laser pulses. Because of the strong two-photon absorption of the PmPV-based films in the mid-infrared (MIR) wavelengths, we used polymethyl methacrylate (PMMA) to host the Bio-Te nanocrystals for our MIR measurements (see Methods). Figure 2e, f present the z-scan results of Bio-Te-PMMA at 2.5 μm and 2.8 μm wavelengths, respectively. The noise level is higher than that in NIR because of the relatively low signal-to-noise ratio of the detector used (a PbS photodiode). While the host PMMA exhibited negligible NLO responses (see Supplementary Fig. 1a), the composite Bio-Te-PMMA film showed obvious saturable absorption at both 2.5 μm and 2.8 μm. This difference implies that the Bio-Te is the only material that contributed to the saturable absorption in Fig. 2e, f. The saturated intensities of Bio-Te-PMMA were fitted as 220 GW·cm$^{-2}$ and 245 GW·cm$^{-2}$ for 2.5 μm and 2.8 μm, respectively. The field of ultrafast mid-infrared fiber lasers is important and active, and these saturable absorptive responses imply that the Bio-Te can serve as a passive mode-locker for such lasers, especially those based on erbium and holmium ions.

We compared the SA responses of Bio-Te with graphene, a well-known NLO material for broadband SA[24–26], over the range of the visible to mid-infrared (MIR) spectrum. In the visible to near-infrared range, we used graphene dispersed in N-Methyl-2-pyrrolidone (graphene-NMP, see Supplementary Note 1) for comparative studies. Compared to the SA response of graphene, the Bio-Te-PmPV response was stronger at 515 nm, comparable at 800 nm, but inferior at 1030 nm (see Supplementary Fig. 2). In the MIR range, we compared tellurium with graphene in Fig. 2e, f with PMMA as the host material (see Methods and Supplementary Note 1), where the Bio-Te exhibited much stronger SA than graphene at both 2.5 μm and 2.8 μm wavelengths. This weak NLO response of graphene may be partially attributed to its two-photon absorption[27], which is more obvious at higher laser irradiance (see the dip of transmission around $z = 0$ cm in

**Table 1 Nonlinear optical coefficients of biological synthesized tellurium**

| Laser | Sample | T (%) | $\alpha_0$ (cm$^{-1}$) | $I_{sat}$ (GW·cm$^{-2}$) | $Im\chi^{(3)}$ (×10$^{-14}$ esu) | FOM (×10$^{-15}$ esu·cm) |
|---|---|---|---|---|---|---|
| 515 nm, 340 fs | Bio-Te-PmPV | 54.0 | 6.17 | 201 ± 35 | −1.07 ± 0.11 | 1.74 ± 0.18 |
| | Graphene-NMP | 46.1 | 7.74 | 364 ± 57 | −0.67 ± 0.09 | 0.86 ± 0.11 |
| 800 nm, 100 fs | Bio-Te-PmPV | 45.0 | 7.99 | 261 ± 176 | −1.2 ± 0.4 | 1.47 ± 0.52 |
| | Graphene-NMP | 16.5 | 18.0 | 910 | −0.968 | 0.54 |
| 1030 nm, 340 fs | Bio-Te-PmPV | 52.4 | 6.47 | 145 ± 23 | −2.76 ± 0.58 | 4.27 ± 0.91 |
| | Graphene-NMP | 52.9 | 6.37 | 56 ± 11 | −5.9 ± 2.1 | 9.3 ± 3.2 |
| 2500 nm, 35 fs | Bio-Te-PMMA | 74.2 | 25.7 | 220 | −20.5 | 7.99 |
| | Graphene-PMMA | 75.7 | 78.9 | 900 | −15.4 | 1.95 |
| 2800 nm, 35 fs | Bio-Te-PMMA | 29.6 | 105 | 245 | −84.1 | 8.03 |
| | Graphene-PMMA | 71.2 | 96.2 | – | – | – |

Saturable absorption coefficients, $I_{sat}$, of Bio-Te and graphene obtained from open-aperture z-scan experiments under various femtosecond laser irradiations, noting that the graphene does not show saturable absorption at 2.8 μm

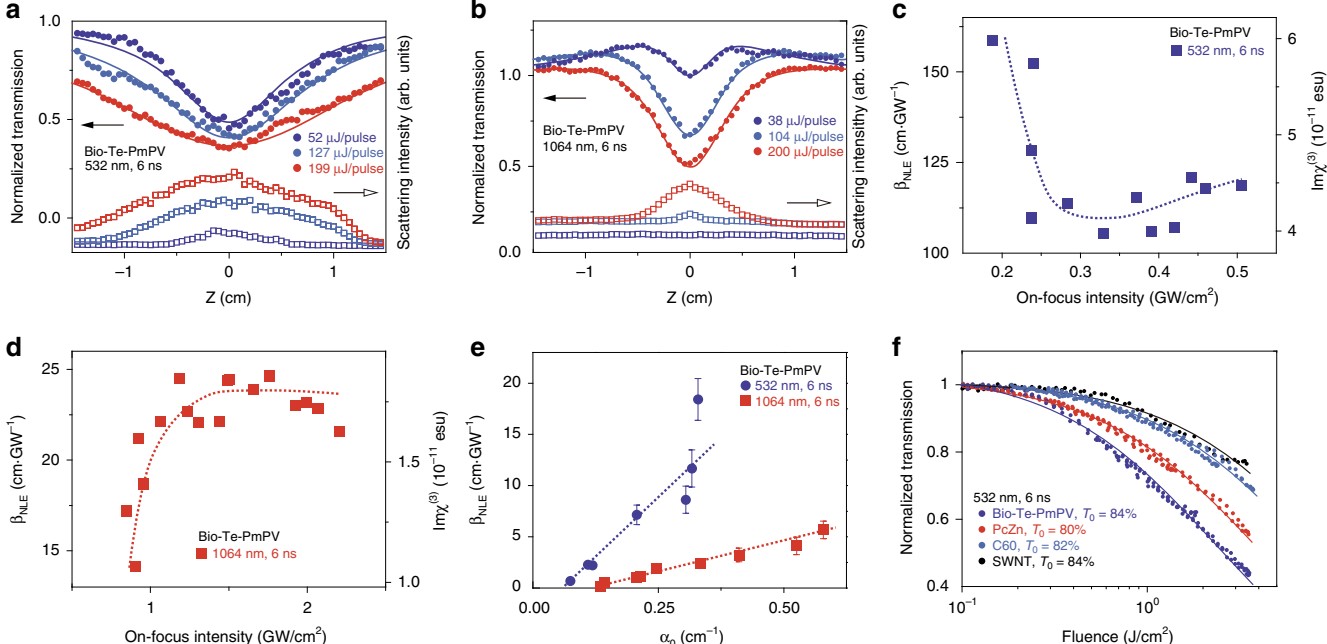

**Fig. 3** Nonlinear optical responses to ns pulses. **a**, **b** Circles: normalized transmission of Bio-Te-PmPV as a function of $z$ at 532 nm and 1064 nm; Squares: light intensity scattered by the sample at 35 degrees to the laser's direction; Lines: z-scan fitting results. **c**, **d** Effective nonlinear extinction (NLE) coefficient $\beta_{NLE}$ and corresponding $Im\chi^{(3)}$ as a function of on-focus intensity for ns pulses at 532/1064 nm. **e** Effective nonlinear extinction coefficient $\beta_{NLE}$ as a function of linear absorption coefficient $\alpha_0$ with error bars indicating s.e.m. **f** Comparison of optical limiting performance of Bio-Te-PmPV, PcZn ($t$-Bu$_4$PcZn), $C_{60}$, and single-walled carbon nanotube (SWNT) dispersions at 532 nm, 6 ns irradiation

Supplementary Fig. 1b). The NLO coefficients of graphene and Bio-Te are presented in Table 1. The strong SA for fs pulses in the infrared region signifies that the biologically synthesized Te holds great technical potential as a passive mode-locker for generating ultra-short laser pulses, which is discussed further later in this paper.

To study the NLO response of Bio-Te-PmPV to nanosecond pulses, we performed z-scans using 6 ns pulses from a 10 Hz Q-switched Nd:YAG laser. The scattered light was collected using a convex lens at ~35° to the direction of the incident beam. The results are shown in Fig. 3a, b. Bio-Te-PmPV exhibited NLE for z-scan traces from 52 μJ to 199 μJ pulse energy at 532 nm (Fig. 3e). Because PmPV only exhibited multi-photon absorption at 532 nm and 1064 nm, no contribution of scattering light would be expected[15]; therefore, the strong scattering signal was attributed to Bio-Te[28]. The absorption of laser pulse energy by Te nanostructures induced micro-bubbles and/or plasmas in the medium, causing strong Mie scattering and hence the attenuation of incident light. The SA effect was minimal in Bio-Te-PmPV (Fig. 3a), which may have been overwhelmed by the much stronger NLE effect. Similar to the fs measurements, we can neglect the effect of PmPV because of its small absorption (0.346 cm$^{-1}$ at 532 nm, with a total absorption coefficient of 3.47 cm$^{-1}$). In this case, the photon transport equation can be simplified as:

$$\frac{dI}{dz} = -(\alpha_0 + \beta_{NLE}I)I \qquad (2)$$

where $\beta_{NLE}$ is the effective NLE coefficient. The fitted z-scan traces according to Eq. (2) corresponded well with the experimental results as shown in Fig. 3a. For 1064 nm (Fig. 3b), the combined effects of SA and NLE were observed in Bio-Te-PmPV. SA was the dominant effect in the low-intensity region, while NLE dominated in the high-intensity regime. In this case, we still employed Eq. (1) for z-scan fitting, and the results are

shown in Fig. 3b. Symmetric valleys around $z = 0$ were observed giving rise to NLE effects; however, $\beta_{NLE}$ is now attributed to prominent nonlinear scattering. The plot of $\beta_{NLE}$ as a function of on-focus intensities for 532 nm and 1064 nm pulses are given in Fig. 3c, d, which may reflect high-order nonlinearities that accompany with the nonlinear scattering in the Te composites. To investigate the relationship between $\beta_{NLE}$ and linear absorptive coefficient $\alpha_0$, we conducted a series of z-scans with different concentrations of Bio-Te dispersions (Fig. 3e). The $\beta_{NLE}$ showed a linear dependence of $\beta_{NLE} = 47.3(\alpha_0 - 0.06)$ (cm·GW$^{-1}$) for 532 nm, and $\beta_{NLE} = 12(\alpha_0 - 0.014)$ (cm·GW$^{-1}$) for 1064 nm.

The broadband NLE exhibited by Bio-Te-PmPV (Fig. 3) can be exploited for laser protective applications[28–30]. To evaluate this potential, systematic comparisons with several well-known NLE materials were carried out (Fig. 3f) which included PcZn ($t$-Bu$_4$PcZn), $C_{60}$ fullerene, and single-walled carbon nanotubes (SWNTs). Additional results comparing Bio-Te-PmPV with graphene and graphene oxide (GO) dispersions[28–32] are shown in Supplementary Fig. 3. Our results demonstrate that Bio-Te-PmPV possesses a much superior optical limiting capability compared to the rest of the other materials under the consideration including graphene and GO, which are considered to be highly promising materials for laser protection. The strong optical limiting capability of Bio-Te-PmPV may benefit from two useful properties of Bio-Te nanoparticles: fast local temperature rises from the large ratio of absorption cross-section $\sigma$ to heat capacity $C_p$, and its low phase-transition melting temperature at 450 °C and boiling temperature at 988 °C ($\sigma/C_p$ of Te is ~4.9 times larger than that of carbon materials such as graphite, and graphite only exhibits sublimation at ~4000 °C. See Supplementary Note 2 for more details). With these properties, the Te nanocrystals can contribute many micro-inhomogeneities (droplets and bubbles) in suspension, thus resulting in strong nonlinear scattering.

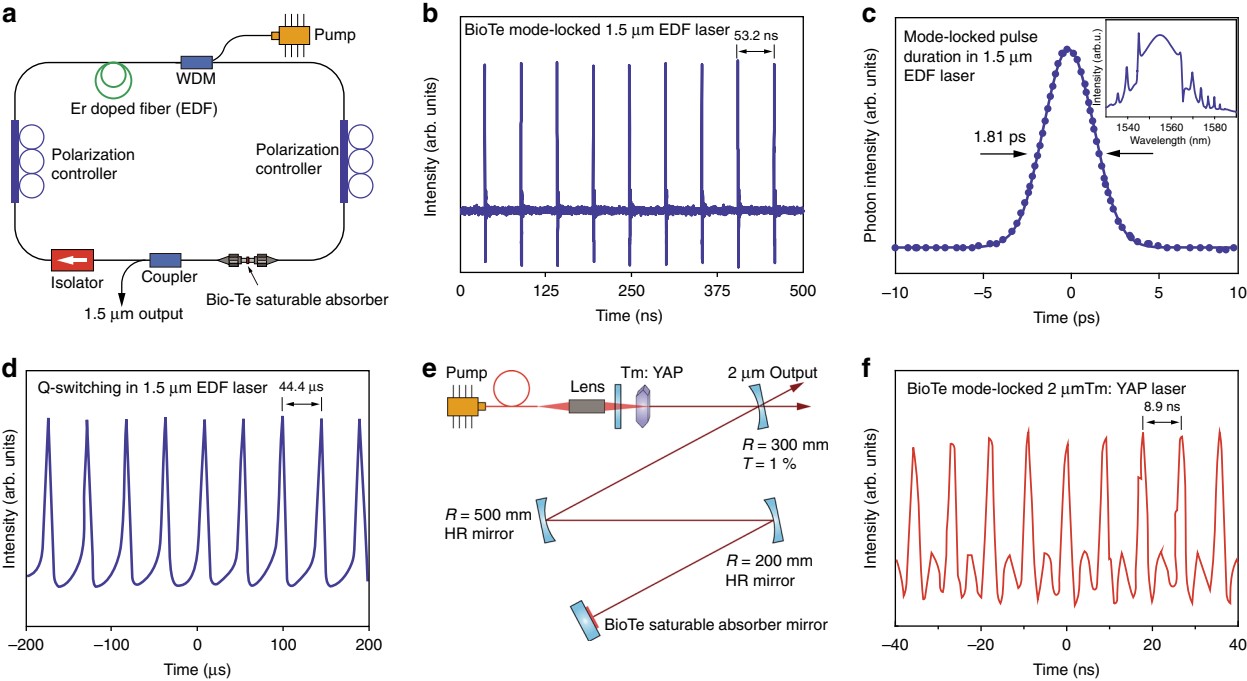

**Fig. 4** Ultrafast laser pulse generation using biologically synthesized tellurium. **a** Scheme of the erbium-doped fiber laser with a Bio-Te-saturable absorber. **b** Mode-locked pulse train generated by the erbium fiber laser with repetition rate of 18.8 MHz. **c** Auto-correlation measurement of Bio-Te mode-locked laser pulses. The sech$^2$ fitting indicates a 1.81 ps pulse width. The inset shows the spectrum of the mode-locked pulses. **d** Q-switched pulse train generated in the EDF fiber laser. **e** Diagram of the Tm: YAP 2 μm laser with Bio-Te saturable absorber mirror acting as a passive optical modulator. **f** Mode-locked pulses generated in the same Tm: YAP laser with the a ~112.3 MHz repetition rate and 1.6 ns pulse duration

A comparison of NLO responses between biologically and commercially available (Sigma Aldrich) chemically synthesized tellurium nanocrystals (Chem-Te) was also carried out. We followed the same procedure to prepare Chem-Te-PmPV composites. As shown in Fig. 1c, Chem-Te and Chem-Te-PmPV have similar Raman fingerprints as their Bio-Te counterparts. The comparison of nonlinear optical performances of Chem-Te and Bio-Te are presented in Supplementary Figs. 2d and 3b. We observed a strong SA in the NIR spectrum in the fs regime and a broadband optical limiting for ns pulses in Chem-Te-PmPV, which were comparable to Bio-Te. These results demonstrate that the microbiologically synthesized Te material has equivalent photonic properties compared to commercially available Te.

**Demonstrations of biological tellurium for photonic applications**. We examined the potential of Bio-Te as a material for photonic uses and demonstrated that Bio-Te nanocrystals can be used for passive mode-locking and Q-switching in infrared lasers near 1.55 μm (Fig. 4a–d). The development of ultra-fast fiber lasers has attracted much attention because of their high stability, large gain, and robust mode confinements[26,33]. To build a Bio-Te ultrafast fiber laser we integrated the Bio-Te nanocrystals into PMMA[34] (see Methods). Bio-Te film was coated on the face of a fiber connector and inserted into the cavity of an erbium-doped fiber (EDF) laser. With the help of the Bio-Te saturable absorber, we observed self-starting mode-locking with 360 mW pump power (Fig. 4b). A repetition rate of 18.8 MHz was achieved corresponding to a cavity length of ~ 10.6 m. The output spectrum of the Bio-Te EDF laser is shown in the inset of Fig. 4c, where few narrow sidebands are distributed along both sides of the 1.55 μm peak. The formation of these Kelly spectral sidebands was due to the periodic

perturbation in the laser cavity, indicating that the laser was operated in the soliton region. Figure 4c also shows the auto-correlation trace of the Bio-Te EDF laser. The pulsed duration was fitted to 1.81 ps with the sech$^2$ function, compared to the 1.46 ps pulse duration obtained for black phosphorus[35]. The maximum power output of ultra-fast pulses was 1.95 mW (see Supplementary Fig. 4), corresponding to an output pulse energy of 103.7 pJ. In addition to our observations on mode-locking, we followed the Q-switching properties of the Bio-Te saturable absorber, as shown in Supplementary Fig. 5. Q-switching was self-started with ~40 mW pump power and disappeared at >110 mW pump power. As the pump power increased from 0 mW to 110 mW, the laser repetition rate increased from 7.5 kHz to 30 kHz, and the pulse width varied from 7 μs to 12 μs (Supplementary Fig. 5). Compared to the 10.5 μs pulse duration in a black phosphorus Q-switching EDF laser with >160 mW pumping[35], our Bio-Te laser could achieve a 7 μs pulse duration with merely 90 mW pumping power, which is a significant reduction in power needs. This comparison implies that Bio-Te is a more promising material than black phosphorus as a Q-switcher. The maximum Q-switched pulse energy of this fiber laser was found to be 10.3 nJ with 100 mW pump power (see Supplementary Fig. 5a, c). The pulse train is shown in Fig. 4d.

Tellurium has a small bandgap of 0.335 eV, comparable to that of black phosphorus[5,22]. This suggests that Bio-Te can be used to generate a mid-infrared ultrafast laser pulse (up to 3.7 μm wavelength)[35,36]. We demonstrated this potential in a Tm:YAP 2 μm solid laser (Fig. 4e). A Bio-Te-PMMA film was coated onto a highly reflective dielectric mirror, thus acting as a saturable absorptive mirror (SAM). We achieved mode-locking using Bio-Te-SAM with a 112.3 MHz repetition rate and a 1.6 ns pulse width (Fig. 4f). Moreover, the Q-switching operation was also demonstrated with maximum pulse energy of 58 μJ and a

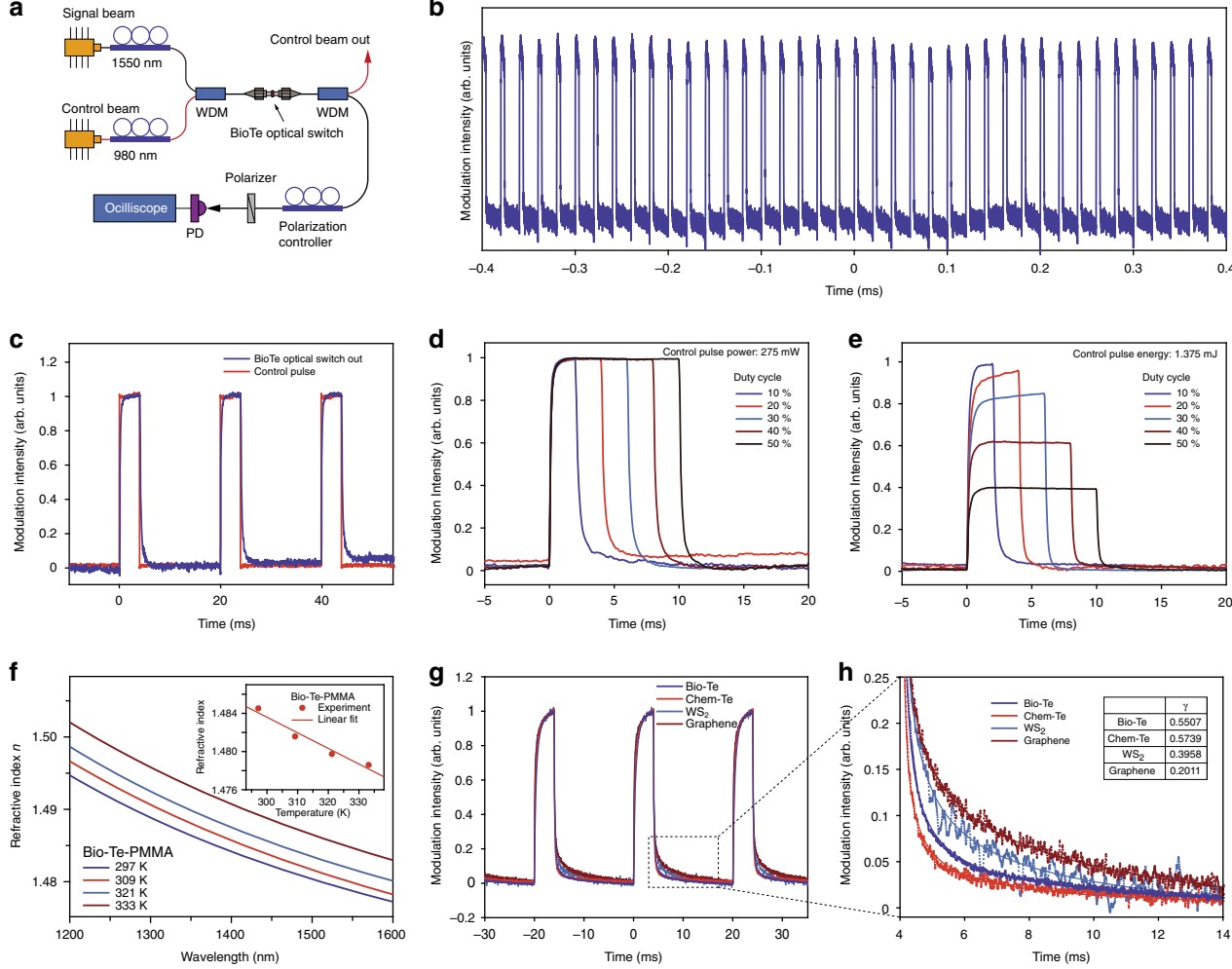

**Fig. 5** Demonstration of all-optical switch based on biologically synthesized tellurium. **a** Scheme for testing the performance of a Bio-Te optical-switch based on polarization interference. **b** Long-term output waveform of the Bio-Te optical-switch, indicating its output stability. The average power and duty cycle of the controlling 980 nm pulses were 55 mW and 20%, respectively. **c** Comparison of the signal input to the output from the Bio-Te optical switch. The rise time and fall time were measured as 276.3 μs and 563.0 μs. **d**, **e** Effect of duty cycles of the control pulse on the output of the Bio-Te optical switch when the control pulse's peak intensity (d) or pulse energy (e) remains constant. **f** Refractive index of Bio-Te-PMMA film measured at different temperatures by an ellipsometer. Inset: the refractive index at 1.55 μm as a function of temperature. **g** Comparison of the output waveforms based on various films serving as working material for optical switching, showing the recovery lifetimes of different materials. **h** Enlarged image from the marked area in (g) showing the performance in signal decay of different materials once the control pulse was turned off. The falling times are determined to be 764 μs and 476 μs for Bio-Te and Chem-Te, and are 1.440 ms and 2.200 ms for WS₂ nanoflakes and graphene, respectively. Solid lines are fitted to the power law, $y \propto t^{-\gamma}$. Inset table: the exponent, $\gamma$, from the fit of the power law

minimum pulse duration of 11 μs (see Supplementary Fig. 6). The maximum power of the Bio-Te Q-switching laser reached 400 mW, suggesting that Bio-Te has a high thermal stability. The versatility of the Bio-Te SA observed from the experimental results suggests that it could be a promising photonic material for mid-infrared lasers.

We investigated the potential application of biological tellurium as an optical switching device in a fiber optic system. All-optical switching is of great importance for signal processing in many applications including optical communication and computation[37,38]. Our Bio-Te optical switch was based on the polarization-dependent thermo-optic effect of Te-PMMA thin film (See Supplementary Note 3). Figure 5a shows the scheme for a Bio-Te optical switch using polarization interference[37]. The 1.55 μm signal beam and 980 nm control beam were generated by two laser sources. We chose this wavelength combination because of the much larger absorption of Bio-Te at

980 nm than at 1.55 μm as shown in Fig. 1f. After the laser sources, two fiber polarization controllers (PCs) were used to adjust the polarization of the signal beam and control beam independently. Both beams were combined using a 0.98/ 1.55 μm wavelength de-multiplexer (WDM) and were channeled into the Bio-Te-PMMA coated fiber connectors. Owing to the thermo-optical effect, Bio-Te introduced polarization-dependent phase shifts in the two orthogonal components of the 1.55 μm signal beam. These two components interfered with each other when they reached the polarizer. This interference created a variable signal output after the polarizer, which was depended upon the power of the control beam.

Figure 5b represents the output of the 1.55 μm pulse train as modulated by the control beam with a 20% duty cycle. Compared with a previous report on the use of WS₂ as an optical switch[39], the Bio-Te optical switch presented a significant advantage in the on/off stability mode. Figure 5c shows the response of the Bio-Te

optical switch, where the rise and fall times were determined to be 276.3 µs and 563.0 µs, respectively, following the 10–90% rule. These response times suggested almost an order-of-magnitude improvement compared to the values reported in similar devices based on other materials: $WS_2$ (rise time of 7.3 ms) as well as graphene optical switches (rise/fall times to be 9.1 ms/3.2 ms)[39,40]. The extinction ratio of the Bio-Te optical switch in Fig. 5c was measured to be ~13 dB. To determine the role of the control pulses, we measured the output waveform of the Bio-Te optical switch as a function of the energy/power of the control pulses. As shown in Fig. 5d, e, the switching time of the signal light depended on the peak power of the pump light but was virtually negligible with the duration time of the control light. These phenomena were consistent with previous experimental measurements reported on other thermo-optic devices[39], implying that the thermo-optic effect was the dominant mechanism in our Bio-Te optical switch.

To obtain the thermo-optic coefficient, we performed a series of temperature-dependent ellipsometry measurements on the Bio-Te-PMMA thin film on a silicon substrate. The results are shown in Fig. 5f, and the refractive indexes at 1.55 µm as a function of temperature are summarized in the inset. The corresponding thermo-optic coefficient of the Bio-Te-PMMA film is fit to be approximately $-1.64 \times 10^{-4} \, K^{-1}$ by linear regression. The thermo-optic coefficient of the Bio-Te nanocrystal was then estimated to be approximately $-3.5 \times 10^{-3} \, K^{-1}$ (see Supplementary Note 3). The previously reported thermos-optic coefficients of graphene and $WS_2$ are $\sim 1.1 \times 10^{-5} \, K^{-1}$ and $\sim 3.4 \times 10^{-4} \, K^{-1}$, respectively[39,40]. The magnitude of the thermo-optic coefficient of Bio-Te is 1–2 orders larger than that of graphene and $WS_2$, implying that Bio-Te can introduce much larger phase changes through the thermo-optic effect than graphene or $WS_2$ with the same thickness.

To experimentally determine which of the materials mentioned above could provide the faster performance, we fabricated various PMMA thin films incorporated with Bio-Te, Chem-Te, graphene, and $WS_2$ flakes, and performed comparative studies on the responses of the thermo-optic switches based on the material used. Figure 5g shows their output waveforms with 20% duty cycle. The extinction ratios were measured to be 13.0 dB, 12.8 dB, 10.6 dB, and 7.5 dB for Bio-Te, Chem-Te, graphene and $WS_2$, respectively. We also studied the response speeds of the thermo-optic switches using different materials. Figure 5f shows the enlarged falling edges of the waveforms. We focused on the falling edges here because they are less steep than the rising edges and the slower processes are more likely to create bottlenecks in high-speed response applications. Employing the power law, $y \propto t^{-\gamma}$, all the falling edges were well fitted (as shown by the solid lines in Fig. 5h). The exponent, $\gamma$, was determined to be 0.551, 0.574, 0.396, and 0.201 for Bio-Te, Chem-Te, $WS_2$, and graphene, respectively, where the larger exponent indicates faster heat dissipation. We determined the respective fall times of the Bio-Te and Chem-Te samples to be 764 µs and 476 µs, while those of the $WS_2$ and graphene samples were 1.440 ms and 2.200 ms. These results suggested that tellurium has obvious advantages over graphene and $WS_2$ for high-speed thermo-optic switching applications.

We successfully harvested biologically generated elemental tellurium nanocrystals (Bio-Te) produced by the anaerobe, *Bacillus selenitireducens*, and studied their potential for nonlinear optical applications that includes optical limiting, mode-locking, Q-switching, and all-optical switching. Bio-Te formed a stable conjugated polymer composite with PmPV, enabling us to construct solid-state devices. We observed a strong saturable absorption in the visible, near-, and mid- infrared optical regimes, indicating that Bio-Te is superior to graphene at 800 nm, 2.5 µm,

and 2.8 µm wavelengths. In addition, Bio-Te-PmPV composites exhibited strong nonlinear extinction responses to nanosecond pulses, which exceeded those of a $tBu_4PcZn$ solution, $C_{60}$ fullerene, single-walled carbon nanotubes, and graphene. By using Bio-Te as a saturable absorber, we successfully generated ultrafast pulse trains by mode-locking/Q-switching in a 1.5 µm erbium-doped fiber laser and a 2 µm Tm: YAP solid-state lasers, with comparable performance to black phosphorus in broadband and short pulse durations. Finally, we demonstrated a superior all-optical switching phenomenon based on the Bio-Te's thermo-optic effects of Bio-Te in optical fiber. This optical switch exhibited excellent performance, with 276.3 µs and 563.0 µs rise and fall times, respectively. The thermo-optic coefficient of Bio-Te was estimated to be approximately $-3.5 \times 10^{-3} \, K^{-1}$, whose magnitude is 1–2 orders greater than those of graphene and $WS_2$. By comparative studies in thermal-optic switching, we proved that Bio-Te provides obvious improvements in the thermal-optic decaying lifetime compared to $WS_2$ and graphene. Our results imply that Bio-Te is a promising bio-material for a broad range of photonic applications.

## Methods

**Synthesis of biological tellurium**. Biological tellurium (Bio-Te) was synthesized by growing haloalkaliphilic anaerobic bacterium, Bacillus Selenitireducens, in a lactate–tellurite medium[9,10]. Briefly recounted, growth of B. Selenitireducens was conducted by pulsed additions of 0.6 mM of Te(IV). The culture was incubated statically at 28 °C for 30 days. During growth Te(0) nano-rods first appeared on the cell surfaces, and then aggregated into larger Te(0) shards. These shards sloughed from the cell and finally accumulated as Te rosettes. To separate the Te(0) rosettes from the bacteria, a lysozyme treatment combined with ultrasonication was employed to break open the bacterial cells. After repeated washing and centrifugation, we obtained assemblages of Te(0) rosettes which were free of cellular debris (Bio-Te). The purified Bio-Te(0) rosettes were stored under a nitrogen atmosphere to preclude oxidation for further experiments.

**Fabrication of Bio-Te-PMMA films**. 1 mL of 4% PMMA-toluene solution was drop-cast into polymer petri dishes (55 mm diameter). After the evaporation of the solvent, PMMA film was formed. Next, another 1 mL of Bio-Te-PmPV solution was drop-cast onto the surface of the PMMA film. After drying at room temperature for ~10 min, the composite film of Bio-Te/PMMA film was obtained.

**Characterization**. The crystal structure of the synthesized Bio-Te was studied using high-resolution transmission electron microscopy (HRTEM) on a FEI Tecnai G220 instrument operating at 200 kV. The sample for TEM study was prepared by dispersing the powder in absolute ethanol followed by drop-casting the solution onto a 400-mesh copper grid. Raman spectroscopy was carried out on a Jobin Yvon LabRam 1B Raman spectrometer with a He: Ne laser operating at 633 nm. Absorption spectra were taken using a PerkinElmer Lambda 750 instrument. PL decay kinetics were measured from a home-built confocal micro-PL spectrometer integrated with an SFM (Catalyst II, Bruker). The samples for PL measurements were fabricated by drop-casting Bio-Te-PmPV in toluene onto cleaned quartz substrates. The thermo-optic coefficients of Bio-Te-PMMA films was measured in a spectroscopic ellipsometer with variable temperature measurement capability (HORIBA Scientific UVISEL 2).

**Z-scans**. An open-aperture Z-scan system was used to study the ultrafast NLO behavior of the Te/PmPV composite. This system measures the total transmittance through a sample as a function of incident laser intensity, while the sample is sequentially moved through the focus of a lens (along the z-axis). For the visible-NIR Z-scans, a mode-locked Ti:Sapphire laser operating at 800 nm with 100 fs pulses at a 1 kHz repetition rate, and a mode-locked fiber laser operating at 1030 nm and 515 nm, 340 fs, 100 Hz repetition rate were employed. For the mid-IR Z-scans we used the idler beam of the Topaz-Prime OPA output pumped by Spitfire Ace laser system (800 nm, 37 fs, 7 mJ pulse energy and 1 kHz repetition rate). The laser pulse energy was determined from average power measured by a thermal power sensor (Thorlabs S401C). A PbS photodetectors (Thorlabs PDA30G) were used to register the incident and transmitted beam signals.

**Bio-Te EDF laser**. The laser cavity was a fiber ring cavity including EDF, 980/1550 WDM, BioTe saturable absorber, polarization controllers, output coupler, and isolator. The EDF (model LIEKKI Er110 8/125) was 0.3-meter length with 110 dB/m absorption at 1530 nm, which was pumped at 980 nm with 720 mW maximum power. The output coupler provided a 10% output ratio. The isolator guaranteed the unidirectional operation of the laser. The entire cavity length was 10.6 m.

## Data availability

All data are available from the corresponding author upon reasonable request.

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

## Acknowledgements

K.P.W. is supported in part at the Technion by a fellowship from the Lady Davis Foundation. J.W. thanks the financial support received from the National Natural Science Foundation of China (NSFC, 61675217, 61875213, 11874370), the Strategic Priority Research Program of CAS (XDB16030700), the Key Research Program of Frontier Science, CAS (QYZDB-SSW-JSC041), and the Program of Shanghai Academic Research Leader (17XD1403900). W.J.B. gratefully acknowledges a visiting professorship for international scientists from the Chinese Academy of Sciences. His contribution towards this publication has also emanated from research supported in part by a research grant from Science Foundation Ireland (SFI) under Grant Number 12/IA/1306. S.M.B. and R.S.O. were supported by the National Research Program of the Water Mission Area of the U.S. Geological Survey. I.M.K. is supported by President's International Fellowship Initiative (PIFI) of CAS (2017VTB0006, 2018VTB0007). Mention of brand-name products does not constitute an endorsement by the USGS. Curran, Liao, and Maharjan would like to acknowledge the assistant from Integricote in funding materials and providing knowhow. The help from Dr. Guohang Hu on ellipsometry measurements is greatly appreciated.

## Author contributions

J.W., W.J.B., and S.A.C. conceive the original idea. J.W., K.P.W. and L.Z. led the project and wrote the paper. S.M.B., K.L., S.M., S.A.C. and R.S.O. cultivated the bacteria and prepared all the nanocrystals. K.P.W., J.W., I.M.K., Q.Z. and G.W. performed the nonlinear optical measurements. J.W., X.Y.Z., and J.F. prepared the dispersions, Te-PMMA films and measured the thermo-optical coefficients. N.D., S.F.Z. and J.F. performed the spectroscopy and the time-resolved PL. H.Z., J.W. and X.Y.Z. preformed the HR-TEM. X.Z., J.D., and Y.L. built the solid-state laser and measured the performance of Q-switching and mode-locking. K.W. and J.C. measured the performance of Bio-Te Q-switched/mode-locked fiber laser and performed thermo-optical switching measurements. K.P.W., J.W., I.M.K. and K.W. contributed the analysis of the data. All authors discussed the results and commented on the paper.

## Additional information

**Competing interests:** The authors declare no competing interests.

