## [Peer Review File · Nature Communications]

Reviewers' comments:

Reviewer #1 (Remarks to the Author):

This manuscript describes the potential use of bacterially grown Tellurium nanocrystal in laser devices. The work focuses mainly on saturable absorption, and to a lesser extent thermo-optic switching effects. The use of bio-grown materials for applications is quite interesting. Given the reported experiments, their potential for use is intriguing, but their real potential remains speculative as described below.

The thermo-optic switching was tested in PMMA-Te composites. Indeed, the authors were able to demonstrate thermo-optic switching. However, their claims of speed are beyond speculative. Given the composite nature, the thermal conductivity will be limited by the polymer host. Indeed, examination of the switching transients indicate that the speed is limited by the thermal conductivity of the polymer in the millisecond range. The authors speculate that high speed switching could be demonstrated. However, they have not listed the thermal conductivity of Te to back this up, nor have they described how they might fabricate a sample that would demonstrate this speed. This would be much more interesting if they would have demonstrated high-speed switching.

Regarding saturable absorption and applications, the material is indeed interesting as a potential high-performance mid-IR saturable absorber. However, experiments are performed in the near-IR, while the authors speculate that the small bandgap promises mid-IR performance. Again, this has not been demonstrated. Its mid-IR performance may be limited because it is formed as a composite and the mid-IR absorption of the host might be a problem. Should the authors fabricate a sample and demonstrate superior mid-IR performance, the work would indeed be of keen interest to the community.

In summary, the use of bio-based Te materials for optoelectronics is quite interesting, and the preliminary experiments suggest that they might be. However, a number of claims are made regarding their potential that are not supported by data in these preliminary experiments. Additionally, the authors do not suggest a path for producing the materials that would demonstrate these capabilities.

The work could be of immediate interest and influence thinking in the field should the authors be able to demonstrate the performance on which they have speculated.

Reviewer #2 (Remarks to the Author):

I refer to the manuscript entitled as "Bacterially Synthesized Tellurium Nanostructures for Broadband Ultrafast Nonlinear Optical Applications" and authored by Wang, et. al.. The authors report a new method to synthesize Tellurium (Te) nanocrystals and present their investigations into the nonlinear optical properties of these Te nanostructures, such as saturable absorption, nonlinear scattering and thermo-optic effects. They also present a number of demonstrations, including mode-locking, Q-switch, optical limiting, and all-optical switch with the as-synthesized Te nanocrystals. Although their results are interesting, there are a number of issues for the authors to consider for the improvement. The major issues are listed below:

Major issues:

- 1) Why the saturable absorption properties of the Te nanocrystals in suspension are superior to other materials such as graphene?

2) Why the nonlinear scattering properties of the Te nanocrystals in suspension are better than other materials such as single walled carbon nanotubes, or PcZn (t-Bu₄PcZn)?

3) In solving Eq. (2) the beta is fixed as a constant; and it is extracted when the numerical solution is fit to the measurement. However, Fig. 3c-d show clearly that the beta is no longer a constant and it depends on the light irradiance. Therefore, it is meaningless. Eq. (2) is wrong in this case.

4) Why the thermo-optic coefficient of the Te composites are larger than the other material mentioned in the manuscript?

5) Can the authors elaborate the positive sign of the thermos-optic coefficient of the Te composites? Why is it?

Minor points:

1) On Page 6, the definition of $\text{Im}\chi$ (3) is proportional to α_{NL} which depends on the light irradiance by $\alpha_{\text{NL}} = \alpha_0 / (1 + I/I_{\text{sat}})$. It is not correct! It should be " $\alpha_{\text{NL}} \approx -\alpha_0 / I_{\text{sat}}$ ".

2) On the caption of Fig. 2: "graphene (red squares) (a)-(c)" should be "Pm PV (red squares) (a)-(c)".

3) On Page 8, just below Equation (2), "Where" should be "where".

4) On the caption of Fig. 3, "nonlinear optical ..." should be "Nonlinear optical ...".

5) On Line 3, Page 12, "155 μm " should be "1.55 μm ".

Reviewers' comments:

Reviewer #1 (Remarks to the Author):

This manuscript describes the potential use of bacterially grown Tellurium nanocrystal in laser devices. The work focuses mainly on saturable absorption, and to a lesser extent thermo-optic switching effects. The use of bio-grown materials for applications is quite interesting. Given the reported experiments, their potential for use is intriguing, but their real potential remains speculative as described below.

Reply: We firstly wish to thank the reviewer for his/her useful suggestions. The reviewer's recognizing on the high quality and well presentation of our work is highly appreciated.

(1) The thermo-optic switching was tested in PMMA-Te composites. Indeed, the authors were able to demonstrate thermo-optic switching. However, their claims of speed are beyond speculative. Given the composite nature, the thermal conductivity will be limited by the polymer host. Indeed, examination of the switching transients indicate that the speed is limited by the thermal conductivity of the polymer in the millisecond range. The authors speculate that high speed switching could be demonstrated. However, they have not listed the thermal conductivity of Te to back this up, nor have they described how they might fabricate a sample that would demonstrate this speed. This would be much more interesting if they would have demonstrated high-speed switching.

Reply: Thanks for the reviewer for his/her suggestion in demonstration of Te as high-speed thermal-optic switching material. To back this up, we have conducted new comparative studies on the thermal-optical responses between a few well-known 2D materials (graphene and WS₂) and tellurium (See Figure R1 below). To this end, we prepared a new bunch of samples with graphene, WS₂ and biological/chemical tellurium hosted in PMMA. The optical switching speeds were investigated by measuring the rise/fall time of the modulated pulses. We found that in our experiments, the rising times have minor difference and are much shorter than the falling times. Therefore, the falling times are the bottle-neck of the response speed in this comparative study. Our measurements show both tellurium sample exhibit obvious less falling time than graphene and WS₂. Moreover, the modulated output from the tellurium samples have better signal-to-noise ratio than those from graphene and WS₂. We believe such experiments demonstrate the tellurium can be better material than graphene and WS₂ for thermal-optical switching. These new results are updated in the Fig 5f-h and the two new paragraphs on Page 15 in revised manuscript.

Figure R1 (a) The comparison of the output signal based on variant films served as working material for optical switching (b) Enlarged image from the shaded area in (g) showing the performance in signal decay of different materials when the control pulse was turned off. The falling times are determined to be 764 μ s, 476 μ s for Bio-Te, Chem-Te, while 1.440 ms and 2.200 ms for WS₂ nanoflakes and graphene. Solid lines are from the fit with power law, $y \propto t^{-\gamma}$. Inset table: the exponent, γ , from the fit of power law.

(2) Regarding saturable absorption and applications, the material is indeed interesting as a potential high-performance mid-IR saturable absorber. However, experiments are performed in the near-IR, while the authors speculate that the small bandgap promises mid-IR performance. Again, this has not been demonstrated. Its mid-IR performance may be limited because it is formed as a composite and the mid-IR absorption of the host might be a problem. Should the authors fabricate a sample and demonstrate superior mid-IR performance, the work would indeed be of keen interest to the community.

Reply: We appreciate for this useful suggestion from the referee. As suggested by the reviewer, we manage to fabricate new samples and demonstrate the saturable absorptive responses at both 2.2 μ m and 2.8 μ m wavelengths. Our new results proved the Bio-Te has obvious saturable absorptive response at 2.2 μ m and 2.8 μ m while the graphene sample for comparison show much weaker nonlinearities at such wavelengths (See Figure R2 below). Graphene is considered as benchmark material for mode-locking [Nat. Photonics 7.11 (2013): 842] and optical modulating [Nat. Photonics 10.4 (2016): 227]. These results directly show the promising mid-IR performance of tellurium over that of graphene. We have updated these points in Fig 2e-f in the revised manuscript. The related discussions can be found in Line 1-13 and Line 19-23 on Page 8.

We thank the referee for his /her suggestion about the further demonstration of mid-IR NLO performance of tellurium, which helps to improve the quality of this work.

Figure R2 Mid-infrared open z-scans of Bio-Te and graphene PMMA films at 2.5 μm (a) and 2.8 μm (b) wavelengths, showing better saturable absorptive responses of Bio-Te than that of graphene.

(3) In summary, the use of bio-based Te materials for optoelectronics is quite interesting, and the preliminary experiments suggest that they might be. However, a number of claims are made regarding their potential that are not supported by data in these preliminary experiments. Additionally, the authors do not suggest a path for producing the materials that would demonstrate these capabilities. The work could be of immediate interest and influence thinking in the field should the authors be able to demonstrate the performance on which they have speculated.

Reply: Following the suggestions by the reviewer, we have carried out additional experiments and demonstrated the capabilities of Te as the saturable absorber in mid-infrared as well as its superior performance in thermal-optical response than WS_2 and graphene. We gratefully appreciate that the referee for his/her perspective suggestions in experiments that we can follow to make this work more complete and impactful.

Reviewer #2 (Remarks to the Author):

I refer to the manuscript entitled as “Bacterially Synthesized Tellurium Nanostructures for Broadband Ultrafast Nonlinear Optical Applications” and authored by Wang, et. al.. The authors report a new method to synthesize Tellurium (Te) nanocrystals and present their investigations into the nonlinear optical properties of these Te nanostructures, such as saturable absorption, nonlinear scattering and thermo-optic effects. They also present a number of demonstrations, including mode-locking, Q-switch, optical limiting, and all-optical switch with the as-synthesized Te nanocrystals. Although their results are interesting, there are a number of issues for the authors to consider for the improvement. The major issues are listed below:

Reply: We thank the reviewer for his/her carefulness to point out our typos and improper expressions that save us from awkward.

Major issues:

- 1) Why the saturable absorption properties of the Te nanocrystals in suspension are superior to other materials such as graphene?

Reply: We thank the referee for his/her consideration on the superior saturable absorption of Te. In visible-NIR range, the better saturable absorption of Te may be due to larger absorption cross-section than that of graphene in some wavelengths or longer saturable absorber recovery time [Keller, U., et al., IEEE J. Sel. Top. Quantum Electron. 2, 435-453 (1996)]. Imaginary part of dielectric permittivity of Te in visible and NIR range is also larger than Si, Ge, and even Au [Science Adv. (2018) 4, 8, 9894]. With $\text{Re}\{\varepsilon\} \cong 18$ and $\text{Im}\{\varepsilon\} \cong 12$ from this Ref. for a Te thin film at $1\ \mu\text{m}$ wavelength we will have the imaginary part of refractive index $k = 2.2$. For monolayer graphene the value $k = 1.82$ at the same wavelength is known [Appl. Phys. Lett. 97, 091904 (2010)]. Considering that molar density of graphene is 3.75 that of Te, we will have their σ ratio of 4.5 in favor of tellurium. In the mid-infrared, the graphene has two-photon absorption, which weakens the saturable response of graphene [Appl. Phys. Lett. 114, 091111 (2019)].

- 2) Why the nonlinear scattering properties of the Te nanocrystals in suspension are better than other materials such as single walled carbon nanotubes, or PcZn (t-Bu4PcZn)?

Reply: We thank the referee for his/her attention on the nonlinear scattering properties of these materials. There are two processes in a nanoparticle suspension that can determine the strength of nonlinear scattering: (1) evaporation of liquid and (2) phase transitions of nanoparticles by laser-induced heat. The evaporation of liquid is that the nanoparticles absorb laser energy and evaporate the surrounding solvent to micro-bubbles. Local temperature rises of a nanoparticle, ΔT , is proportional to the ratio of its absorption cross-section σ to heat capacity C_p , i.e., $\Delta T \propto \sigma/C_p$ [I.M. Kislyakov et al., Opt. Express, 26(26), p. 34346; J. Chem. Phys. 97(11), 8748–8759 (1992)]. The σ/C_p of Te to that of graphite, $(\sigma/C_p)_{\text{Te}} / (\sigma/C_p)_{\text{graphite}}$, is estimated to be ~ 4.87 (See below about the calculation details). This gives that Te nanoparticles will heat up ~ 4.9 times more at the same irradiance applied. The phase transitions of nanoparticles can also generate microbubbles by themselves. For Te nanocrystals, it has two phase-transitions: melting at 450°C and boiling at 988°C . For carbon materials, however, it only exhibits sublimation at a much higher temperature up to $\sim 4000^\circ\text{C}$ at ambient pressure [W. M. Haynes, ed., *CRC Handbook of Chemistry and Physics*, 97th ed. (CRC, 2017)]. Therefore, for optical limiting, the Te nanocrystals can contribute much more micro-inhomogeneities (droplets and bubbles) than carbon materials because of Te phase transitions. Such discussions are important and we thank the referee for reminding us. We emphasize this point in the revised manuscript (Line 21 Page 10 to Line 5 Page 11)

As for the comparison with PcZn, from literature [Opt. Lett. 19, 625-627 (1994); Int. Rev. Phys. Chem. (2012) 31(3), 319.] we know that in the optical limiting behavior of PcZn (also for other phthalocyanine solution) either two-photon absorption or reverse saturable absorption are prevailing.

These effects are determined solely by properties of the molecular electronic structure and represent a different physics to the nonlinear scattering processes we discussed above.

Calculation of σ/C_p for Te and graphite:

The linear absorptive coefficient α_0 has a relationship $\alpha_0 = N\sigma \cong \rho\sigma N_A/M$, where the N is absorber particle density, σ is absorptive cross-section, ρ is material density, N_A is the Avogadro constant and M is the molar mass. From literature [CRC Handbook of Chemistry and Physics, 97th ed.], we know that the absorption coefficient α_0 of graphite $\sim 2.21 \times 10^3 \text{ cm}^{-1}$ and tellurium $\sim 8.43 \times 10^3 \text{ cm}^{-1}$ at 1 eV photon energy. The molar thermal capacity C_p of Te and graphite is 25.73 J/mol/K and 8.517 J/mol/K respectively. The molar mass, M , of Te and graphite is 127 g/mol and 12 g/mol. The density ρ of Te and graphite is 6.3 g/cm³ and 2.3g/cm³. With all the data above, the $(\sigma/C_p)_{\text{Te}}$ / $(\sigma/C_p)_{\text{graphite}}$ is calculated to be ~ 4.87 .

3) In solving Eq. (2) the beta is fixed as a constant; and it is extracted when the numerical solution is fit to the measurement. However, Fig. 3c-d show clearly that the beta is no longer a constant and it depends on the light irradiance. Therefore, it is meaningless. Eq. (2) is wrong in this case.

Reply: We thank the author for attentions on the correctness of Eq.(2). The Eq. 2, $dI/dz = -(\alpha_0 + \beta_{\text{NLE}})I$, as its form shows, takes account into second-order nonlinearity, i.e., nonlinearity with order higher than 3 won't be reflected by this equation. This equation has shown great fits to the experimental z-scan curves in Fig. 3a-b, implying the beta exacted from these fittings are reasonable for the corresponding laser pulse energy. The reason that the beta changes with the light irradiance is likely due to higher order nonlinearities, which didn't show in a single z-scan curve in Fig. 3a-b. The net influences of such high order effects between individual z-scans on the same sample can appear as a constant add up to beta. By tracking beta obtained with different laser irradiance, we can track the high-order nonlinearities and this is exactly what we showed in Fig 3c-d. To show this point, we add the sentence "These variations of β_{NLE} as laser irradiance may origin from complex high-order nonlinearities accompany with the nonlinear scattering in the Te composites." in Line 1-2 Page 10 in the revised manuscript.

The referee may have question that why we don't directly use high order equations (like $dI/dz = -(\alpha_0 I + \beta I^2 + \gamma I^3 + \dots)$) to catch such high order nonlinearities. The reason is we don't know how much orders we should include in the model and add up more unknow terms would deteriorate fitting quality rapidly if the high-order effect is not obvious. Therefore, as long as the less-order model gives reasonable fitting, we would keep the model as simple as possible. Similar methods can be seen in literatures [such as the Fig. 7 in Adv. Mater. 15 (1), 19-32 (2003)] and this is very useful the nonlinear mechanisms are complex, especially for nonlinear scattering of Te in this work. Therefore, we believe the use of Eq. 2 is appropriate in this manuscript.

4) Why the thermo-optic coefficient of the Te composites are larger than the other material mentioned in the manuscript?

Reply: Thanks for consideration of the larger thermo-optic coefficient of Te composite. This is an open question need to be carefully discussed. Thermal expansion and temperature-dependent polarizability are the two major mechanisms that effect the thermo-optic coefficient of materials [Lit 1]. The thermo-optic coefficient induced thermal expansion are usually negative sign and not strongly depends on the wavelengths, while that induced by polarizability should be positive sign and wavelength-dependent [Lit 2].

From our new results from temperature-dependent ellipsometry (See Fig R3 in the response to Comment 5), the refractive index change of Te is a negative function of temperature and weakly depend on the wavelength from 1.2 μm to 1.6 μm . Thus, thermal expansion is likely to be the major effect in the thermo-optic coefficient of Te. Moreover, the thermal expansion coefficients of Te, graphene and WS_2 are reported to be $29.7 \times 10^{-6} \text{ K}^{-1}$, $-8.0 \times 10^{-6} \text{ K}^{-1}$ and $7-10 \times 10^{-6} \text{ K}^{-1}$ [Lit 3-5], where we can see Te has obviously larger thermal expansion than other materials mentioned in the manuscript. Therefore, we believe the large thermal expansion coefficient may explain the strong thermo-optic effect of Te.

Literatures:

- (1) M. E. Thomas, Handbook of Optical Constants II, edited by E. D. Palik, (Academic, New York, 1991). Page 177-199.
- (2) GE Jellison Jr et al., Journal of Applied Physics 76, 3758 (1994)
- (3) H Ibach, E Ruin and Phys. Stat. Sol. 41, 719 (1970).
- (4) D Yoon, YW Son and H Cheong, Nano Lett. 11, 3227-3231 (2011).
- (5) K.M. McCreary et al., Scientific Reports 6, 35154 (2016).

5) Can the authors elaborate the positive sign of the thermos-optic coefficient of the Te composites? Why is it?

Reply: Thank you for considering the sign of thermo-optic coefficient of Te composites and helping us to find this mistake. The sign should be negative and we correct this in revised manuscript. To confirm the value of thermo-optic coefficient, we have performed a series of temperature-dependent ellipsometry measurement on the Bio-Te-PMMA thin film on silicon substrate. The results are showed in the Figure R3 (a) below. The refractive indexes at 1.55 μm as a function of temperature are summarized in Figure R3(b), where the corresponding thermo-optic coefficient of Bio-Te-PMMA thin film is fit to be about $-1.64 \times 10^{-4} \text{ K}^{-1}$.

To obtain the approximate thermo-optic coefficient of Bio-Te nanocrystal, we did the following estimation: The contribution is approximately proportional to the ratio of optical intensity distribution and in a thin film this is the ratio of material volume [K Wu et al., Photonic Research,

6, 22-28 (2018)] In our Bio-Te-PMMA film, the volume ratio of Bio-Te : PMMA is ~1:100. Then the contribution to the refractive index change can be approximately given by $\Delta n \cong 1\% \cdot \Delta n_{Te} + 99\% \cdot \Delta n_{PMMA}$. Knowing that the thermo-optic coefficient of pure PMMA is $-1.3 \times 10^{-4}/K$ [Z. Zhang et al., Polymer, 47,4893-4896 (2006)]. The thermo-optic coefficient can be estimated to be $-3.5 \times 10^{-3}/K$. We have updated these points in revised manuscript. (Fig. 5f and Line 1-10 Page 15)

Figure R3 (a) Refractive index measured at different temperature by ellipsometer. (b) Refractive index at 1553.69 nm at different temperature.

Minor points:

1) On Page 6, the definition of $Im\chi^{(3)}$ is proportional to α_{NL} which depends on the light irradiance by $\alpha_{NL} = \alpha_0 / (1 + I/I_{sat})$. It is not correct! It should be “ $\alpha_{NL} \approx -\alpha_0 / I_{sat}$ ”.

Reply: This comment is important, we gratefully thank the referee for pointing this out. We have fixed this error and corresponding $Im\chi^{(3)}$ displayed in Table 1.

2) On the caption of Fig. 2: “graphene (red squares) (a)-(c)” should be “Pm PV (red squares) (a)-(c)”.

Reply: All the typos mentioned above have been fixed as suggested.

3) On Page 8, just below Equation (2), “Where” should be “where”.

Reply: Thanks for pointing this out. Now it is fixed.

4) On the caption of Fig. 3, “nonlinear optical ...” should be “Nonlinear optical ...”.

Reply: The caption of Fig. 3 has been capitalized now.

5) On Line 3, Page 12, “155 um” should be “1.55 um”.

Reply: This typo has been corrected as suggested. We are grateful to the referee’s carefulness for pointing out our errors and typos to improve the quality of this manuscript.

REVIEWERS' COMMENTS:

Reviewer #1 (Remarks to the Author):

The authors have adequately addressed the concerns expressed in the first review. The English needs some polishing, but I leave that to the editorial staff. It is now suitable for publication.

Reviewer #2 (Remarks to the Author):

The authors have addressed my comments. Now, I recommend it to be published.

REVIEWERS' COMMENTS:

Reviewer #1 (Remarks to the Author):

The authors have adequately addressed the concerns expressed in the first review. The English needs some polishing, but I leave that to the editorial staff. It is now suitable for publication.

Reply: We thank you for suggesting to polish the language. We have improved the English in the secondly revised manuscript.

Reviewer #2 (Remarks to the Author):

The authors have addressed my comments. Now, I recommend it to be published.

Reply: We appreciate the referee's positive feedback. Thank you very much.